# Joint Random Forest and Particle Swarm Optimization for Predictive Pathloss Modeling of Wireless Signals from Cellular Networks

Okiemute Roberts Omasheye [1], Samuel Azi [2], Joseph Isabona [3], Agbotiname Lucky Imoize [4,5],
Chun-Ta Li [6,7,*] and Cheng-Chi Lee [8,9]

1 Department of Physics, Delta State College of Education, Mosogar 331101, Nigeria
2 Department of Physics, University of Benin, Benin City 300103, Nigeria
3 Department of Physics, Federal University Lokoja, Lokoja 260101, Nigeria
4 Department of Electrical and Electronics Engineering, Faculty of Engineering, University of Lagos, Akoka, Lagos 100213, Nigeria
5 Department of Electrical Engineering and Information Technology, Institute of Digital Communication, Ruhr University, 44801 Bochum, Germany
6 Program of Artificial Intelligence and Information Security, Fu Jen Catholic University, New Taipei City 24206, Taiwan
7 Department of Information Management, Tainan University of Technology, Tainan City 71002, Taiwan
8 Research and Development Center for Physical Education, Health, and Information Technology, Department of Library and Information Science, Fu Jen Catholic University, New Taipei City 24206, Taiwan
9 Department of Computer Science and Information Engineering, Asia University, Taichung City 41354, Taiwan
* Correspondence: th0040@mail.tut.edu.tw

**Abstract:** The accurate and reliable predictive estimation of signal attenuation loss is of prime importance in radio resource management. During wireless network design and planning, a reliable path loss model is required for optimal predictive estimation of the received signal strength, coverage, quality, and signal interference-to-noise ratio. A set of trees (100) on the target measured data was employed to determine the most informative and important subset of features, which were in turn employed as input data to the Particle Swarm (PS) model for predictive path loss analysis. The proposed Random Forest (RF-PS) based model exhibited optimal precision performance in the real-time prognostic analysis of measured path loss over operational 4G LTE networks in Nigeria. The relative performance of the proposed RF-PS model was compared to the standard PS and hybrid radial basis function-particle swarm optimization (RBF-PS) algorithm for benchmarking. Generally, results indicate that the proposed RF-PS model gave better prediction accuracy than the standard PS and RBF-PS models across the investigated environments. The projected hybrid model would find useful applications in path loss modeling in related wireless propagation environments.

**Keywords:** path loss measurement; signal strength intensity; particle swarm optimization; random forest; hybrid RF-PS model; wireless network modeling

## 1. Introduction

Information dissemination in cellular communication system channels is in form of Electromagnetic waves (EM) [1–3]. The signal intensity of the EM reduces as the separation distance between the receiving antenna and the backbone transmitting antenna increases [4–6]. Signal strength reduction is largely caused by the different varying EM propagation mechanisms, such as scattering, diffraction, absorption, etc., impacting the signals during propagation [7–9]. The resultant effect of these complex environmental phenomena is described as a signal path loss [2,10]. Path loss quantifies the degree of signal attenuation between the mobile antenna and the transmitting antenna through space [11]. During cellular network planning and deployment, accurate path loss assessment and prediction help to effectively determine the field intensity of the propagated

signal, coverage quality and the signal interference-to-noise ratio [12]. In the existing literature, several generic path loss models for signal coverage and path loss prediction have been proposed [7,8]. However, these generic models always have precision problems. Thus, without calibrating these models, they can hardly be utilized effectively in practice to produce reliable results, especially in other related environments [13,14].

The linear least square (LLS) regression [15] method has been engaged to calibrate popular path loss models for enhanced prediction of measured path loss obtained in Irbid, Malaysia, Oman, Nigeria, China, and Baghdad city, among others [13,14,16–22]. Particularly, a recursive algorithm, which is an improved version of the LLS approach was employed to adjust the Okumura–Hata model over an operational CMDA system network [14]. In [16], a statistical LLS-based method was explored to tune the Hata model for path loss prediction in Baghdad City. In [17], a Minimax LLS algorithm was explored for automatic calibration of the Ericsson path loss model. An adaptive LLS method which uses a polynomial function for path loss model adjustment has been presented [18]. A similar method for adjusting the Erceg path loss model was presented [19]. The absolute least square regression procedure was explored to optimize the existing path loss model for robust predictive analysis [23]. The major problem with the LLS path loss model calibration and their extended/modified approaches is sensitivity to outliers on measured signal strength, resulting in large signal prediction errors [24,25]. A few works have employed a nonlinear least square approach based on different numerical integration schemes to calibrate existing path loss models for optimal prediction [26,27].

Most recently, the adaptive application of machine learning models, such as neural networks [28–30], support vector machines [31], genetic algorithms [32], and Gaussian process [33] among others, have gained ground toward efficient path loss predictions. While these machine learning models are good models for prediction analysis, they are, however, computationally demanding and inefficient when engaged for stochastic datasets with high dimensions [34]. In [35], a hybrid structure which combines a ray tracing scheme with random forest (RF) was explored for signal attenuation prediction. The proposed hybrid path loss model showed enhanced prediction compared to the preliminary methods. It is interesting to note that similar hybrid schemes jointly combining RF with Radial Basis Function (RBF) and Neural Network (NN) with RBF have been explored for evapotranspiration and path loss prediction [36,37].

In the literature [38–40], population-based models, such as particle swarms and genetic algorithms, have been proposed for path loss predictive modeling. Although the population-based modeling approach showed enhanced predictions, their overall performance could be limited when employed in high-dimensional input features, especially when the predictor number is far larger than the observation number [41]. Though datasets with high-dimensional input features provide more information, the redundant and irrelevant constituents of the data may lower the prediction accuracy of the model [42]. In the current contribution, Random Forest-Particle Swarm (RF-PS) optimization is proposed for adaptive modeling and predictive analysis of signal path loss with high-dimensional input features obtained from 4G cellular networks in the mid-southern part of Nigeria. The relative performance of the proposed RF-PS-based model has been compared to the standard PS and hybrid RBF-PS approach for benchmarking purposes. To this end, the vital contributions of this paper include:

- Measurement-based acquisition of detailed signal data and computation of attenuation loss levels across selected urban LTE microcellular radio communication paths using professional TEMS investigation tools.
- Effective application of the random forest technique for the most informative and important subset of features selection from measured signal data sets toward robust predictive analysis.
- Development and application of an improved signal path loss model using hybrid random forest and particle swarm optimization for optimal cellular planning across the investigated locations.

- Validation of the developed signal path loss models in other eNodeB (base stations) across the investigated locations to ascertain the level of their prediction accuracies.

The remaining parts of this paper are organized as follows. Section 2 describes the methods and materials. Section 3 presents the results and useful discussions. Finally, the conclusion to the paper is given in Section 4.

## 2. Methods and Materials

The materials and methods are briefed in this section. The data collection method is described and the theory covering signal propagation through free space is presented. The Random Forest (RF) [43–46] and Particle Swarm Optimization (PSO) [38,47,48] methods are described as well. Additionally, the Hybrid RF-PS path loss modeling, Radial Basis Function (RBF) networks [49], the proposed hybrid path loss prediction modeling approach and performance indicators are described briefly.

### 2.1. Data Collection

The measurement campaign was carried out in four different cities in southern Nigeria. The cities include Agbor and Asaba in Delta State and Onitsha and Awka in Anambra State. A total of 16 eNodeB were monitored, four in each of the four cities categorized as locations 1, 2, 3, and 4. A TEMS software installed in a laptop with GPS and TEMS handset connected to it was housed in a test vehicle. The TEMS software was launched on the laptop after which the relevant Nigerian routes and cell reference details were uploaded. The TEMS was locked on a 4G LTE network at the 2600 MHz frequency band. Following this, a drive test was conducted on the specified routes in and around the coverage area of the base stations and the reference signal received power was continuously recorded with the TEMS tools. The GPS data were recorded simultaneously at all instants of the collected field data. The Reference Signal Received Power (RSRP) of the serving base stations was recorded on a log file by the TEMS software with detailed information, such as the coordinates of the mobile station location, transmitting frequency, cell identity, distance covered, altitude, and other system parameters. The log file containing the measured RSRP in all the studied locations was saved for further processing and analysis.

### 2.2. Signal Propagation through Free Space

The free space path loss model [40] provides details or means of quantifying the loss that can be attained when radio signals are propagated without considering the effects of several other external impediments in the propagation paths. Thus, the power density $S_p$ [2,40,50], attained over a communication distance $r$ in free space, is related to the received power $P_r$, the transmit power $P_t$, and the antenna gain $G_t$, given by (1), and the received power is given by (2)

$$S_p = \frac{P_t G_t}{4\pi r^2} \tag{1}$$

$$P_r = S_p A_e = \frac{P_t G_t}{4\pi r^2} A_e \tag{2}$$

where $A_e$ is the antenna aperture area defined in (3)

$$A_e = \frac{G_r \lambda^2}{4\pi} \tag{3}$$

$P_r$ can be rewritten with respect to $A_e$ as in (4):

$$P_r = \frac{P_t G_t}{4\pi r^2} A_e = \frac{\lambda^2}{(4\pi r)^2} G_t P_t G_r \tag{4}$$

where $\lambda$ = transmission wavelength in meters.

Thus, the path loss, $P_l(dB)$ over the free space channel can be computed as in (5) using Equation (4):

$$P_l(dB) = 20 \, log\left(\frac{4\pi r}{\lambda}\right) = 20 \, log\left(\frac{4\pi f_{ca} r}{C_{speed}}\right) \tag{5}$$

where $C_{speed}$ and $f_{ca}$ define the speed of light and transmission frequency, respectively.

Given that $C_{speed} = 3 \times 10^8$ m/s, $\pi = 3.142$, then Equation (5) can be expressed as (6):

$$P_l(dB) = 130 + 20 \, log(f) + 20 \, log(r) \tag{6}$$

The expression in (6) reveals that signal loss in free space attenuates 20 dB in value. However, this is certainly not true in other propagation environments, such as built-up terrains, a suburban or an urban area.

In a more general form, Equation (6) can be rewritten as (7):

$$P_l(dB) = a_1 + a_2 \, log(f) + a_3 \, log(r) \tag{7}$$

Thus, Equation (7) is referred to as the general log-distance path loss model. In literature, several efforts have been explored to determine the loss coefficients: $a_1$, $a_2$, and $a_3$. In this paper, these loss coefficients represent the identified parameters to be tuned (optimized) to reflect the true nature of the terrain where the signal is being propagated. This quest has also led to the introduction of several empirical models, such as the Hata [51], SUI [52,53], COST 231 Hata [54–56], and ITU-R M2412-0 [57] models. The application of these models for path loss estimation in an environment different from environments where the models were developed produces significant errors. The results of the optimized loss coefficient (i.e., the identified parameters) of Equation (7) using the proposed RF-PS method compared to other popular existing methods are reported in this paper.

### 2.3. Random Forest

Random Forest (RF), known as an ensemble of the decision tree, is a distinctive non-parametric supervised machine learning method proposed by Breiman and Cutler [46]. RF employs several multiple decision trees to handle regression, feature selection and classification problems. The RF remain an efficient tool for preprocessing datasets via dimensionality reduction or redundancy reduction. Though datasets with high-dimensional input features provide more information, the redundant and irrelevant constituents may lower the prediction accuracy. In this paper, the RF method was employed to extract more relevant features from the measured signal dataset, while eliminating the redundant and less important ones. The RF algorithms have two or more main hyperparameters, which must be provided before engaging them for data training or regression analysis [42]. One such hyperparameter is the number of trees. Mathematically, the RF input-output function model is defined in (8).

$$RF(x_n, y_n) = \{f(x_n, \theta_m, y_n)\} \tag{8}$$

where $\theta_m$ indicates the tree number. The $x_n, y_n$ indicate the input and target output data. Here, to accomplish the task, a set of trees (100) were engaged on the target measured signal data sets to determine the most informative and important subset of features.

### 2.4. Particle Swarm Optimization (PSO)

Particle swarm optimization (PSO) is a population of particle search optimization algorithms introduced by Kennedy and Eberhart in 1995 [47]. In the PSO, populations of particles move in a particular search space at different velocities and randomly choose the candidate solutions. Accordingly, the position of a particle set defines the solution to the optimization problem. The particles search for the best positions in the search space iteratively (step-wise repeated manner) by changing the velocities per the guiding

rules inspired by the flocking bird behavioral model. In each iteration, the velocities and positions can be expressed (9) and (10):

$$V_i^{q+1} = \omega V_i^q + g_1 r_1 (P_{best} - X_i^q) + g_2 r_2 (G_{best} - X_i^q) \tag{9}$$

$$X_i^{q+1} = X_i^q + V_i^{q+1} \tag{10}$$

where $q$ = iteration number, $\omega$ = weight parameter, $r_2$ = coefficient social parameter, $r_1$ = cognition parameter, $X_i^q$ = individual position at iteration $q$, $V_i^q$ = individual velocity at iteration $q$, and $P_{best}$ = best value of each particle.

*2.5. Hybrid RF-PS Path Loss Modelling*

This section reveals how the proposed hybrid path loss model development technique is accomplished. The proposed hybrid path loss model termed RF-PS combines random forest and particle swarm optimization methods. The output of the RF signal is fed into the particle swarm component for further prognostic modeling.

In line with the principle of mean square minimization (MSE), the objective function of the proposed hybrid path loss model can be articulated as (11) and (12):

$$J(\boldsymbol{a}) = \frac{1}{N} \sum_{i=1}^N [y(k) - \overline{y}(k)]^2 \tag{11}$$

$$J(\boldsymbol{a}) = \frac{1}{N} \sum_{i=1}^N [y(k) - \overline{y}(\boldsymbol{a}, x)]^2 \tag{12}$$

where: $N$ = number of path loss data sample, while $y(k)$ = measured path loss data sample $\overline{y}(k) = f(\boldsymbol{a}, x)$ = target prediction response. The problem of accurately determining the modelling parameters, $\boldsymbol{a} = [a_1, a_2, a_3]$, entails minimizing the objective function, $J(\boldsymbol{a})$ expressed in equation (12). The above problem can be tagged as a minimization optimization problem with constraints: min $J(\boldsymbol{a})$ simplified in (13)

$$J(\boldsymbol{a}) = \left\langle \begin{array}{c} a_{1,min} \leq a_1 \leq a_{1,max} \\ a_{2,min} \leq a_2 \leq a_{2,max} \\ a_{3,min} \leq a_3 \leq a_{3,max} \end{array} \right\rangle \tag{13}$$

To implement the PSO technique to solve the minimization optimization problem, let $a_1^{q,p}, a_2^{q,p}, a_3^{q,p}$ designate the solution generated by PSO for q iteration and p population. Thus, the target prediction response can be expressed as (14). The predicted solution can be achieved utilizing Algorithm 1.

$$y = a_1^{q,p} + a_2^{q,p} * log_{10} x + a_3^{q,p} \, log_{10} f \tag{14}$$

---

**Algorithm 1: PSO implementation Pseudocode**

---

1: initialization
2: **Input**:
   Set variables number, n and swarm size
    Set of initial parameters
     Objective function $J(\boldsymbol{a})$, $a = (a_1, a_2, a_3)$
3: **Output**:
   Set of best initial parameters
    Prediction model, $y_{pred}$
4: Start PSO
5:   for $i$ = 1:s do
6:    Evaluate $J(\boldsymbol{a})$;
7:     $P_{best} := \boldsymbol{a}_i$;
8:  end for
9: while (Halt condition) do
10:   Compute the inertia weight $w_i$
11:    for $i$ = 1:s do
12:    if Xi. $X_{max}$ then
13:     Xi = Xmax;
14:   end if
15:     if Xi\Xmin then
16:    Xi = Xmin;
17:   end if
18: Appraise J(**a**)
19:    if $J(\boldsymbol{a})\backslash J(P_{best})$ then
20:     pBesti : = Xi;
21:   end if
22:     if $J(pBesti)\backslash J(G_{best})$ then
23:      $G_{Best}: =P_{Best};$
24:   end if

---

*2.6. Radial Basis Function (RBF) Networks*

In computational neural network modeling, Radial Basis Function (RBF) networks, introduced by Broomhead and Lowe in 1988 [58], represent a distinctive type of feed-forward network with robust universal curve fitting approximation capabilities in a high dimensional space. Accordingly, the RBF networks are generally trained to map input vectors, $x_i \in R^i$, into output vectors, $y_i \in R^0$, where the sets $(x_i, \; y_i)$, $1 \leq i \leq n$, form the training pairs. Thus, learning is the same as determining a surface in the high dimensional space that delivers the best line of fit to the training dataset.

In terms of the radial basis architecture shown in Figure 1, the RBF networks are characteristically made of three layers: one linear output layer, one hidden layer which utilizes a specific non-linear RBF transfer function, and an input layer. The neurons in the input layer have a linear function which primarily feeds and links the input signal vector to the hidden layer. The input layer outputs are obtained by computing the distance between hidden layer centers and the network inputs. Generally, the RBF networks can be expressed mathematically as [59]:

$$y_j(\mathbf{x}) = w_{jo} + \left(\sum_{i=1}^{n} w_{ji}\varphi_j(\mathbf{x})\right) \tag{15}$$

$i\in\{1,2,\ldots, n\}, j\in\{1,2,\ldots, J\}$

The expression in Equation (15) can be simplified further by introducing an additional basis function, $\phi_0 = 1$, and this yields (16):

$$y_j(\mathbf{x}) = \sum_{i=0}^{n} w_{ji}\phi_j(\mathbf{x}) \tag{16}$$

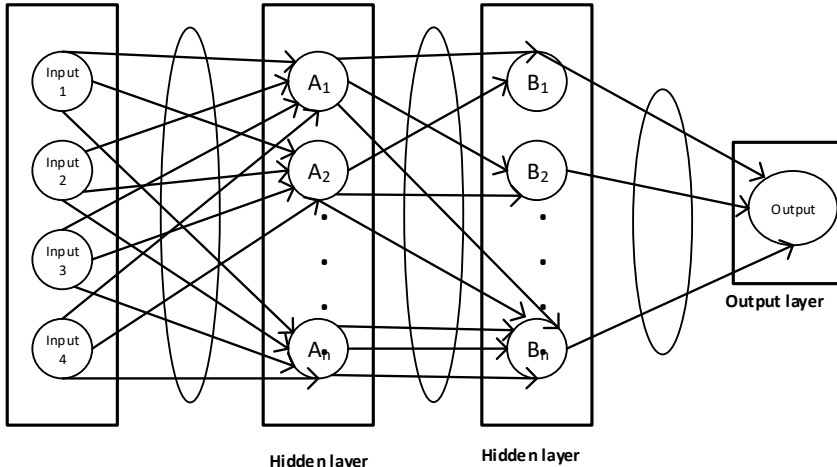

**Figure 1.** Radial basis architecture.

In terms of Gaussian basis functions, $\phi_j(\mathbf{x})$ is defined by (17):

$$\phi_j(\mathbf{x}) = \exp\left(-\frac{\|\mathbf{x} - \boldsymbol{\mu_j}\|^2}{2\sigma_j^2}\right) \tag{17}$$

where $y_j$ = network output of the $j$th neuron, $n$ = no. of hidden layer neurons, $J$ = dimension of the output, $\phi$ = radial basis function, $W_{ji}$ = weight of the $j$th output and $i$th neuron, $\sigma$ = spread parameter for the $i$th neuron, $\mathbf{x}$ = input data vector, and $\boldsymbol{\mu_j}$ = center vector of the $i$th neuron.

In the literature [36,37], the RBF network has been combined with particle swam (PS) optimization as a hybrid approach for optimal predictive analysis. Thus, in this paper, the RBF is introduced in combination with PS to enable us to benchmark the proposed hybrid RF-PS method.

### 2.7. The Proposed Hybrid Path Loss Prediction Modeling Approach

The proposed hybrid path loss prediction model which combines the Random Forest (RF) and Particle Swarm Optimization (PSO) algorithm is briefly summarized. The hybrid approach is tagged as the RF-PS method. To develop, implement, and test the RF-PS model performance, three-phase steps are exploited, as illustrated in Figure 2.

*i.* Study Locations Field Survey and Signal Strength Measurement*:* This phase consists of the study locations, field survey, detailed signal strength measurements, signal extraction, and preliminary preprocessing.

*ii.* Path Loss Calculation and Analysis based on Measured Signal Strength*:* This consists of path loss analysis in comparison with some key existing path loss models.

*iii.* Hybrid Path Loss Prediction Modeling and Application*:* To achieve this crucial task, the spatial signal data is first passed through RF for dimensionality and feature selection process. A set of trees (100) on the target measured signal data sets were considered to determine the most informative and important subset of features, which were in turn employed as input data to the particle swarm optimization model for predictive path loss analysis.

The proposed hybrid path loss model combines the random forest and particle swarm optimization methods, termed RF-PS. Thirdly, detailed validation, application of the RF-PS model, and comparison with other existing techniques are presented.

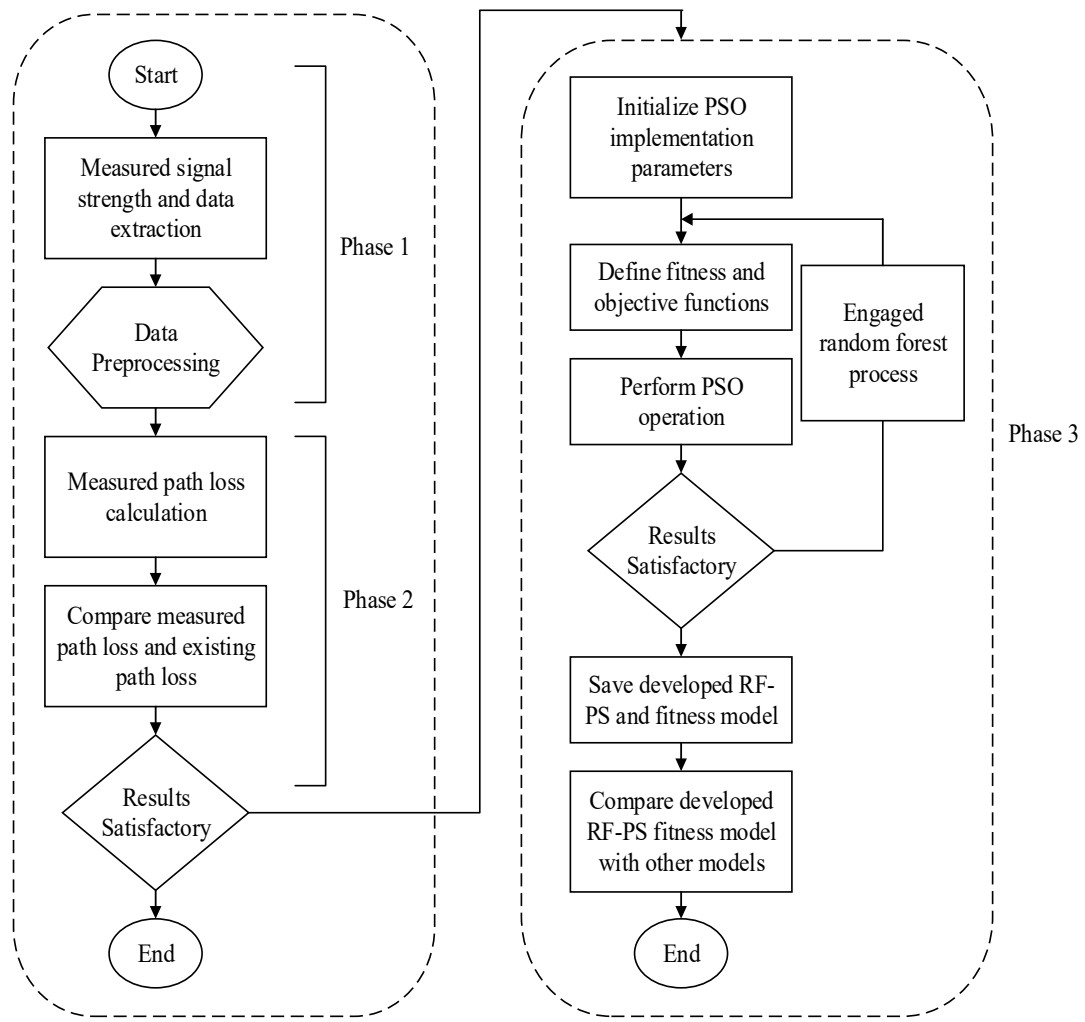

**Figure 2.** A three-phase flow chart for the proposed RF-PS path model development and application.

*2.8. Performance Index*

Three key performance indexes were employed to statistically quantify and examine the prediction accuracy of the proposed RF-PS model. These include Mean Absolute Error (MAE), Root Mean Square Error (RMSE), and Correlation Coefficient (R) [60–62]. The mathematical definitions of these performance indexes are given in Equations (17)–(19). Specifically, the Mean Absolute Error (MAE) is given in (18), the (RMSE) is shown in (19), and the (R) is given in (20)

$$MAE = \frac{1}{N} \sum_{q=1}^{N} \left| y_q - d_q \right| \tag{18}$$

$$RMSE = \sqrt{MSE} = \frac{1}{N} \sqrt{\sum_{q=1}^{N} \left[ y_q - d_q \right]^2} \tag{19}$$

$$R = \frac{\sum\limits_{q=1}^{N} \left( y_q - \overline{y}_k \right) \left( y_q - \overline{d}_k \right)}{\sqrt{\left[ \sum\limits_{q=1}^{N} \left[ (y_k - \overline{y}_k)^2 \right] \right] \left[ \sum\limits_{q=1}^{N} \left[ \left( y_k - \overline{d}_k \right)^2 \right] \right]}} \tag{20}$$

where $y_q$ denotes the desired target output, $d_q$ indicates the actual network output, $\bar{y}_q$ is the mean of the actual network output, $q = 1, 2, \ldots, N$, and $N$ is the actual network output number.

## 3. Results and Discussions

The results of this study and valuable discussions are presented. The results are divided into four parts in line with the objectives of the study. The first part concentrates on the quantification and analysis of the measured path loss in comparison with the standard path loss models. The second part presents the enhanced prediction using the developed hybrid Random Forest and Particle Swarm (RF-PS) optimization method over the standard path loss optimization methods. The third part provides detailed results using the developed (RF-PS) optimization method for path loss prediction in other study locations used for model validation.

### 3.1. Quantification and Analysis of the Measured Path Loss in Comparison with the Standard Path Loss Models

There is a need to quantify and compare measured path loss in comparison with the standard path loss models before employing the proposed RF-PS method for path loss model optimization. Thus, this first part concentrates on the quantification and analysis of the measured path loss in comparison with the standard path loss models such as the COST 231 Hata and ITU-R M.2412-0 models. As shown in Figures 3–6, the graphs reveal the level of practical measured signal path loss as a function of measurement distances across the four study locations. The locations include: Asaba, Onitsha, Awka, and Agbor. In each of the aforementioned study locations, one eNodeB site each was engaged to obtain the path loss data. From the graphs, the levels of the measured path loss in Asaba, Onitsha, Awka, and Agbor vary between: 90–144 dB, 100–150 dB, 110–140 dB and 106–120 dB, respectively. For the standard path loss models, the COST 231 Hata and ITU-R M.2412-0 models vary between 200 and 240 dB and between 171 and 175 dB, respectively. From the results, it is noticeably clear that loss values attained by the standard models are quite higher than the ones obtained through field measurements. In terms of accuracy, the results also imply that the standard COST 231 Hata and ITU-R M.2412-0 models over predicted the measured path loss values across the four study locations. As a case in point, the COST 231 model achieved 78.27, 83.75, 78.27 and 82.77 dB RMSE values, and the ITU-R M.2412-0 model gives 27.29, 34.39, 64, 28, 32.76 dB values, respectively. The higher path loss values produced by the standard models may be ascribed to the physical terrain and topographical differences between the actual locations where the measurement was conducted and the terrain characterization where the models were initially developed. Thus, the need to fine-tune the existing standard models for reliable and improved prediction of the measured path loss data is self-evident and worthy of investigation.

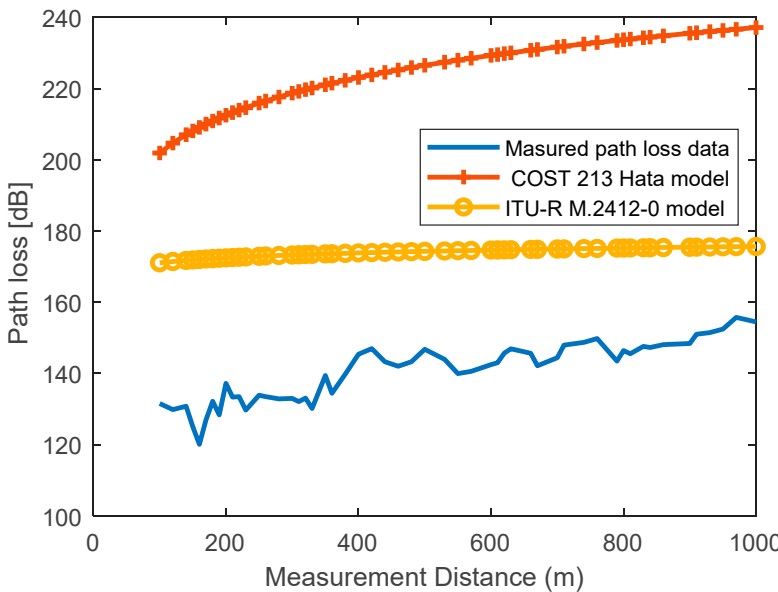

**Figure 3.** Measured Path loss values versus distance and Prediction comparison with standard COST 231 and ITU-R M.2412-0 model in site 1 Location 1.

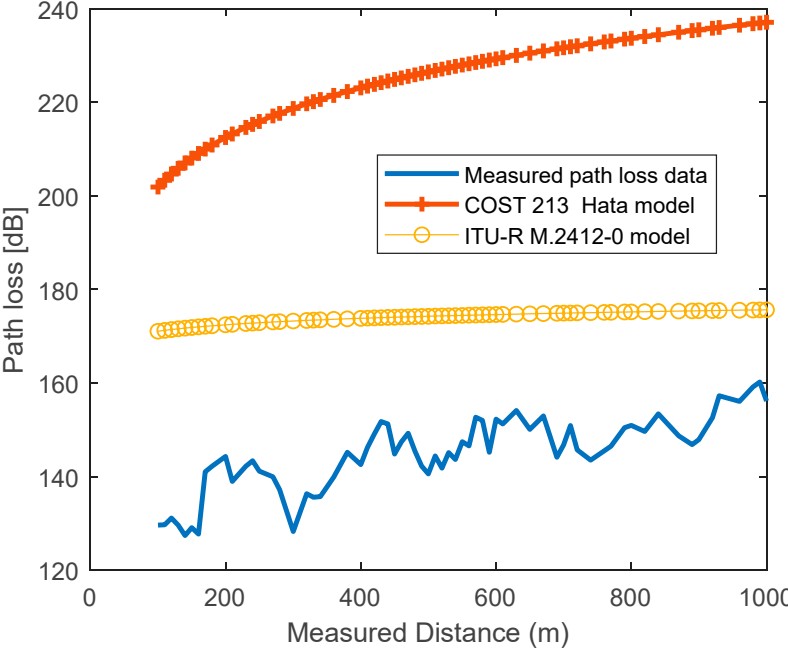

**Figure 4.** Measured Path loss values versus distance and Prediction comparison with standard COST 231 and ITU-R M.2412-0 model in site 1 Location 2.

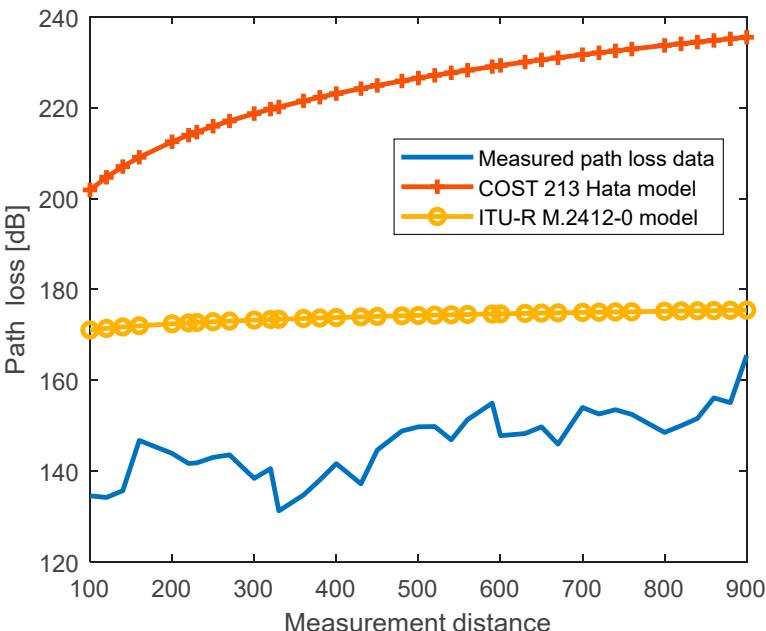

**Figure 5.** Measured Path loss values versus distance and Prediction comparison with standard COST 231 and ITU-R M.2412-0 model in site 1 Location 3.

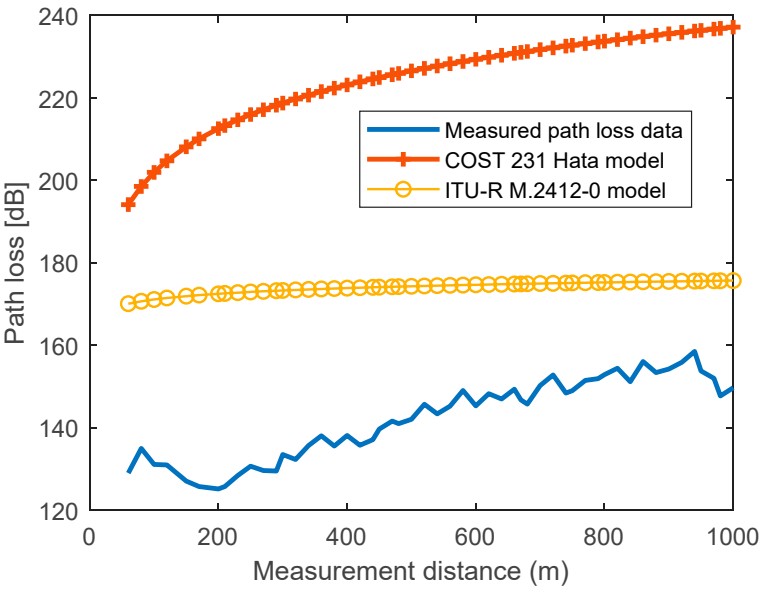

**Figure 6.** Measured Path loss values versus distance and Prediction comparison with standard COST 231 and ITU-R M.2412-0 model in site 1 Location 4.

### 3.2. Dimensionality Reduction through Important Data Feature Selection Using RF

A set of trees (100) on the target measured signal data sets was employed to determine the most informative and important subset of features, which were in turn employed as input data to the particle swarm optimization model for predictive path loss analysis. Table 1 and Figure 7 show the acquired signal dataset with constituent features in one of the studied locations and the number of features selected via the methods are displayed.

**Table 1.** The acquired signal dataset and the consistent features in one of the studied locations.

| BTS LAT. | BTS LONG. | CELL ID | DIST. (m) | RSRP | RSRP LAT. | RSRP LONG. | FREQ. (MHz) | ALT (m) | Path Loss (dB) |
|---|---|---|---|---|---|---|---|---|---|
| 6.2497 | 6.2022 | 297 | 100 | −82.5 | 6.2504 | 6.2028 | 2600 | 170 | 81.3000 |
| 6.2497 | 6.2022 | 297 | 120 | −68.81 | 6.2506 | 6.2028 | 2600 | 169 | 82.3300 |
| 6.2497 | 6.2022 | 297 | 140 | −70.19 | 6.2508 | 6.2028 | 2600 | 169 | 97.5000 |
| 6.2497 | 6.2022 | 297 | 160 | −89.25 | 6.2509 | 6.2028 | 2600 | 169 | 83.8100 |
| 6.2497 | 6.2022 | 297 | 180 | −75.75 | 6.2512 | 6.2028 | 2600 | 169 | 85.1900 |
| 6.2497 | 6.2022 | 297 | 200 | −70.44 | 6.2515 | 6.2029 | 2600 | 169 | 104.2500 |

x1 = BTS Latitude, x2 = BTS Longitude, x3 = Cell ID, x4 = Distance (m), x5 = RSRP, x6 = RSRP Latitude, x7 = RSRP Longitude, x8 = Frequency, x9 = Altitude.

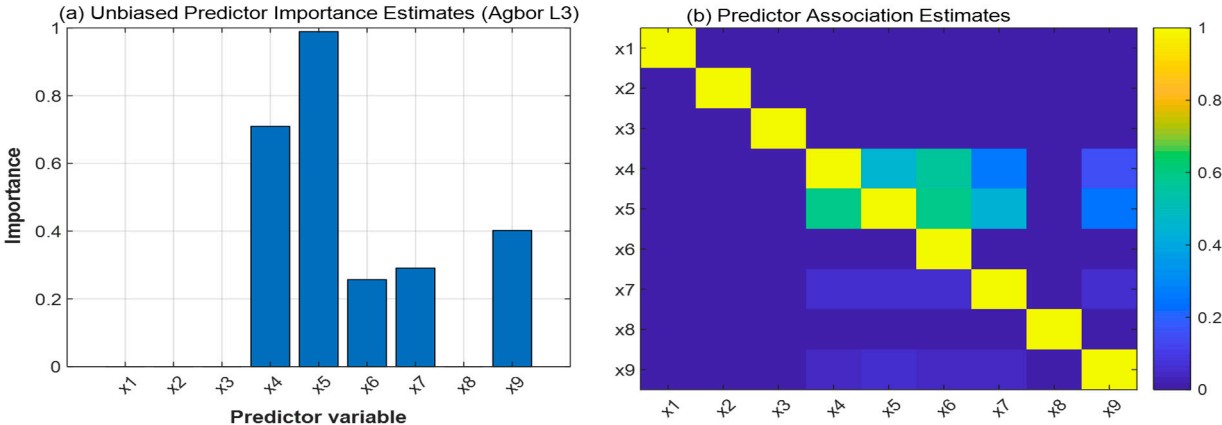

**Figure 7.** The constituent feature (predictor) selection of the important estimates and their association using RF. (**a**) Unbiased predictor importance estimates for Agbor L3. (**b**) Predictor association estimates.

Figure 7a reveals the level of importance each measured signal dataset feature attained, and Figure 7b displays each constituent feature (predictor) association estimate. A bigger estimated value indicates a more important constituent feature. As shown in Figure 7a, the RSRP predictor has the highest level of importance with a score of 1. This score is followed by distance with a 0.7 score. Moreover, RSRP, Latitude, and Altitude (m) scored 0.28, 0.3, and 0.4, respectively. On the other hand, BTS Latitude, BTS Longitude, Cell ID, and frequency scored zero each, meaning they have no impact on predictions.

### 3.3. Enhanced Prediction Obtained with the Developed Hybrid RF-PS Optimization Method over the Standard Path Loss Optimization Methods

The results obtained by applying the proposed hybrid RF-PS optimization method over the standard path loss optimization methods are presented in Figures 8–19. In order to benchmark, the ordinary PS and RBF-PS methods used for path loss model optimization are used for performance analysis across the studied locations. Three key performance indicators: RMSE, MAE, and R, are used in this contribution. The proposed RF-PS optimization method showed better prediction performance with lower RMSE values; 3.51–4.33 dB, 2.55–2.98 dB, 3.23–3.54 dB, and 3.11–5.22 dB for locations 1–4, respectively. For standard PS and RBF-PS optimization methods, the RMSEs are quite higher; 3.96–6.08 dB, 2.69–4.33 dB, 4.60–6.38 dB, and 4.160–6.16 dB for locations 1–4, respectively. Overall, the robustness and superiority of the proposed RF-PS method for path loss prediction over the existing techniques are demonstrated.

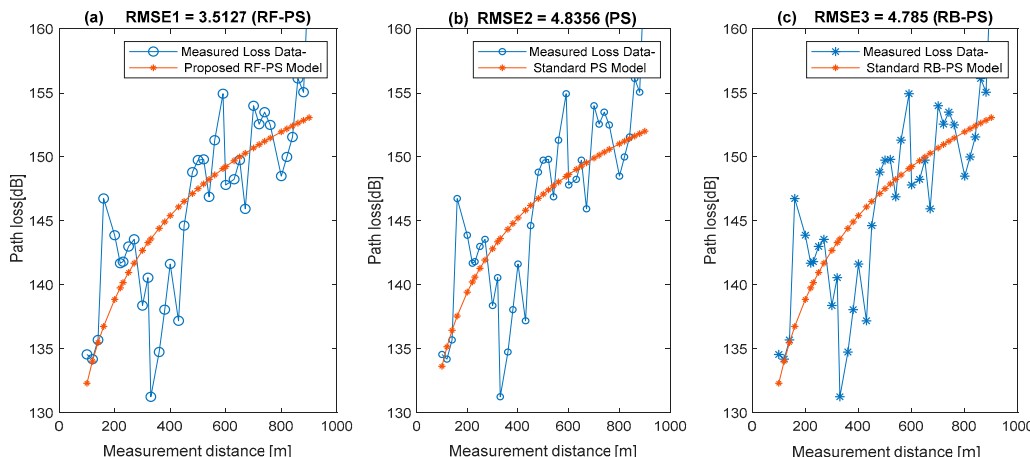

**Figure 8.** Results showing the measured path loss versus distance using the hybrid RF-PS method and two existing standard path loss prediction methods in site 1 of location 1.

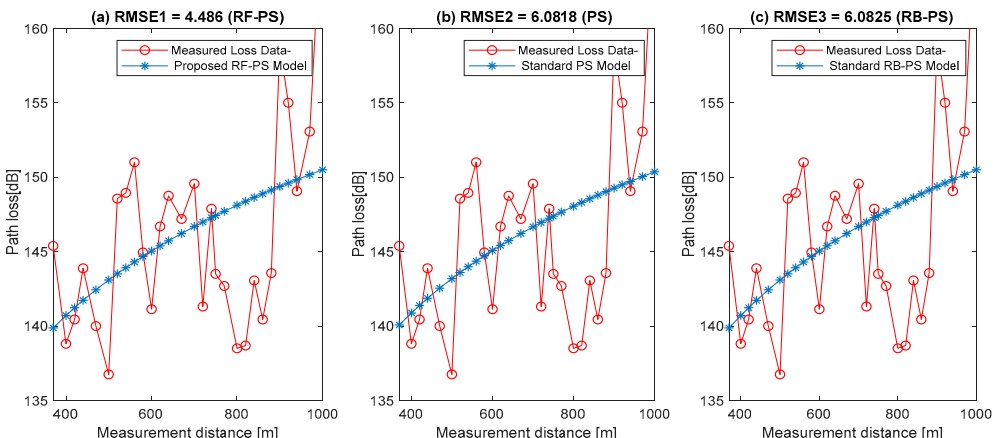

**Figure 9.** Results showing the measured path loss versus distance using the hybrid RF-PS method and two existing standard path loss prediction methods in site 2 of location 1.

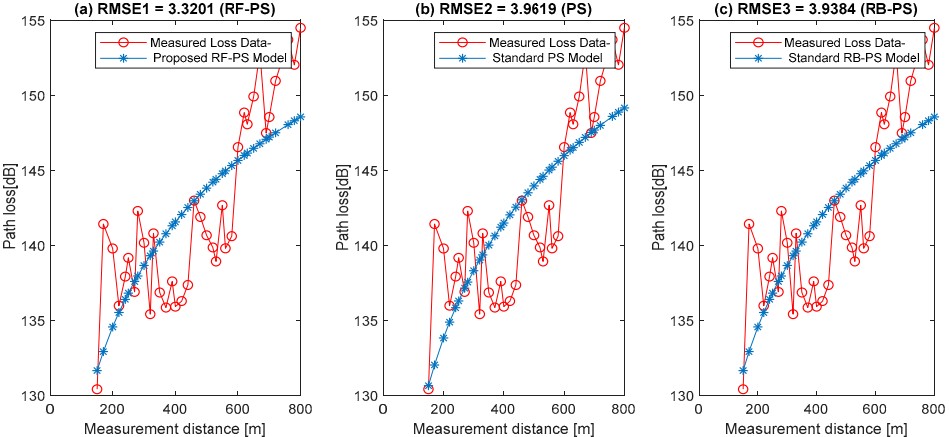

**Figure 10.** Results showing the measured path loss versus distance using the hybrid RF-PS method and two existing standard path loss prediction methods in site 3 of location 1.

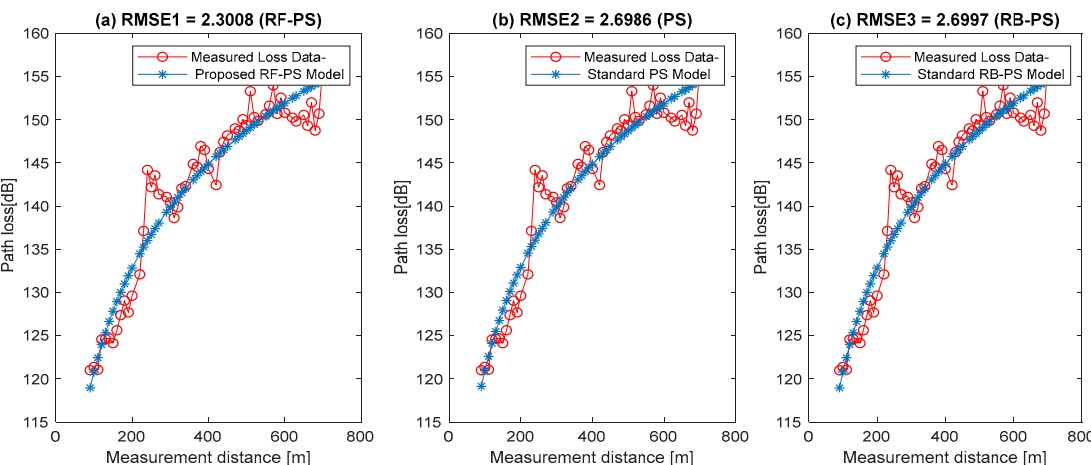

**Figure 11.** Results showing the measured path loss versus distance using the hybrid RF-PS method and two existing standard path loss prediction methods in site 1 of location 2.

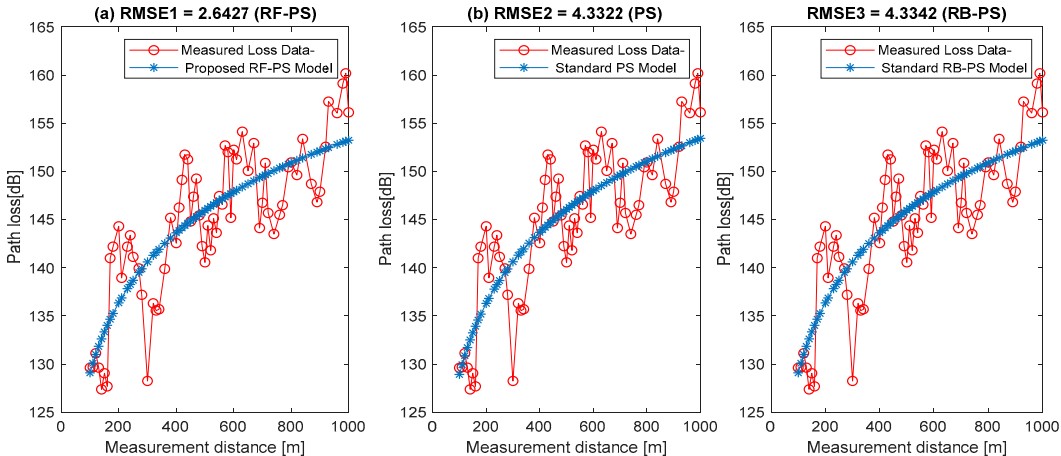

**Figure 12.** Results showing the measured path loss versus distance using the hybrid RF-PS method and two existing standard path loss prediction methods in site 2 of location 2.

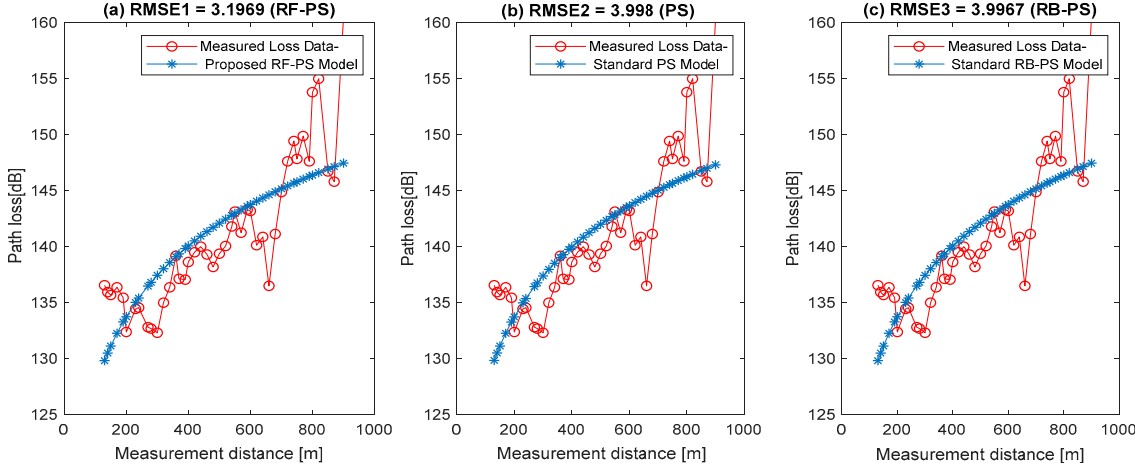

**Figure 13.** Results showing the measured path loss versus distance using the hybrid RF-PS method and two existing standard path loss prediction methods in site 3 of location 2.

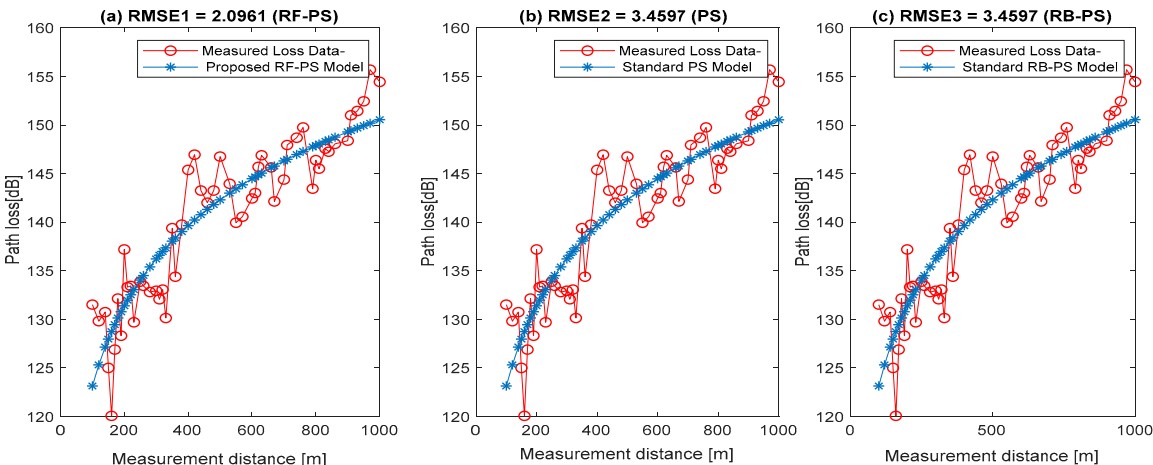

**Figure 14.** Results showing the measured path loss versus distance using the hybrid RF-PS method and two existing standard path loss prediction methods in site 1 of location 3.

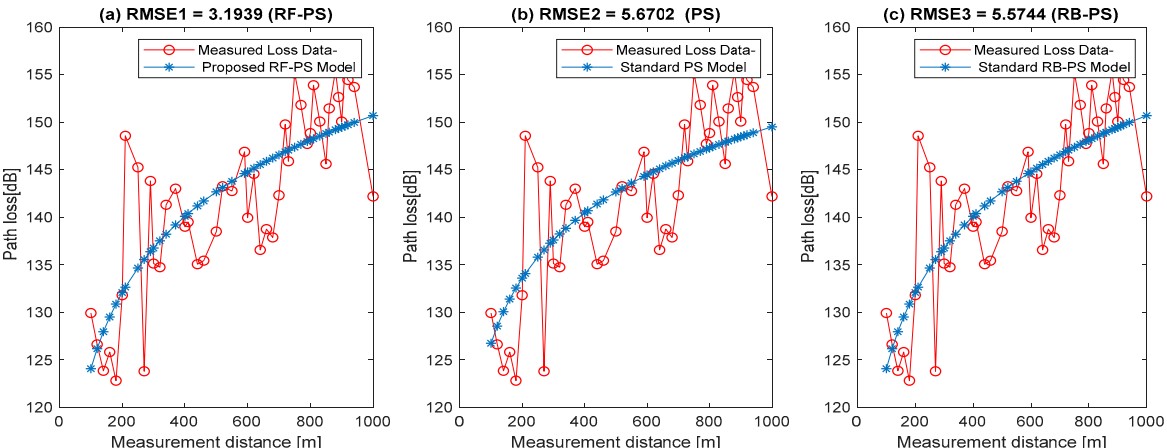

**Figure 15.** Results showing the measured path loss versus distance using the hybrid RF-PS method and two existing standard path loss prediction methods in site 2 of location 3.

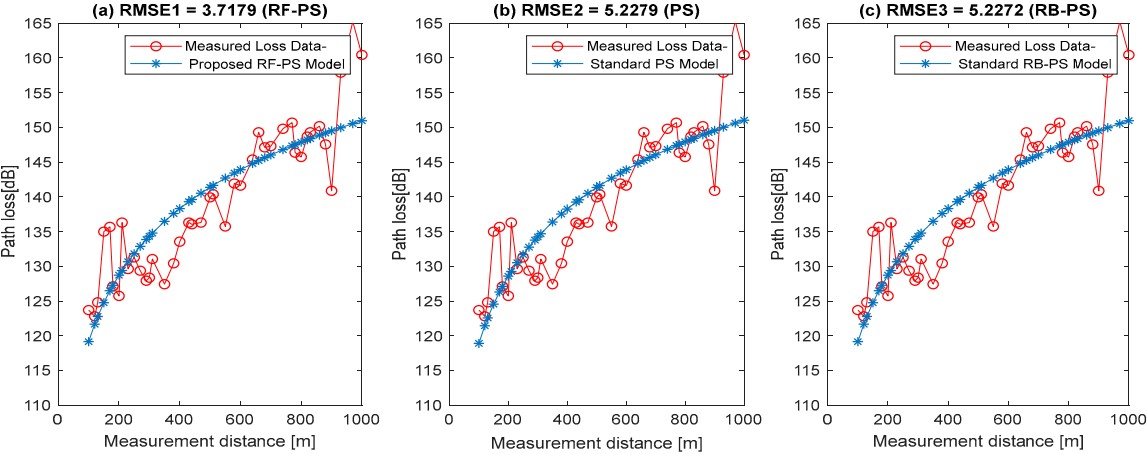

**Figure 16.** Results showing the measured path loss versus distance using the hybrid RF-PS method and two existing standard path loss prediction methods in site 3 of location 3.

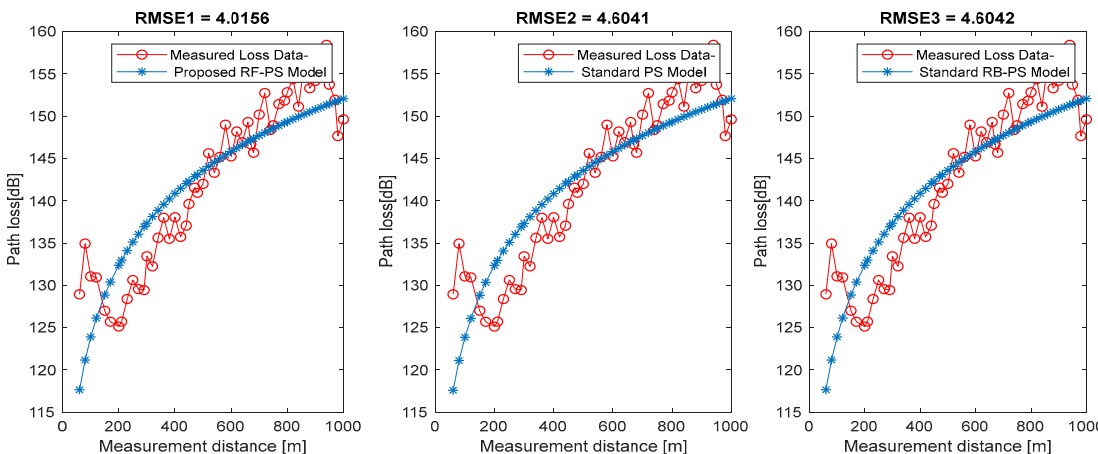

**Figure 17.** Results showing the measured path loss versus distance using the hybrid RF-PS method and two existing standard path loss prediction methods in site 1 of location 4.

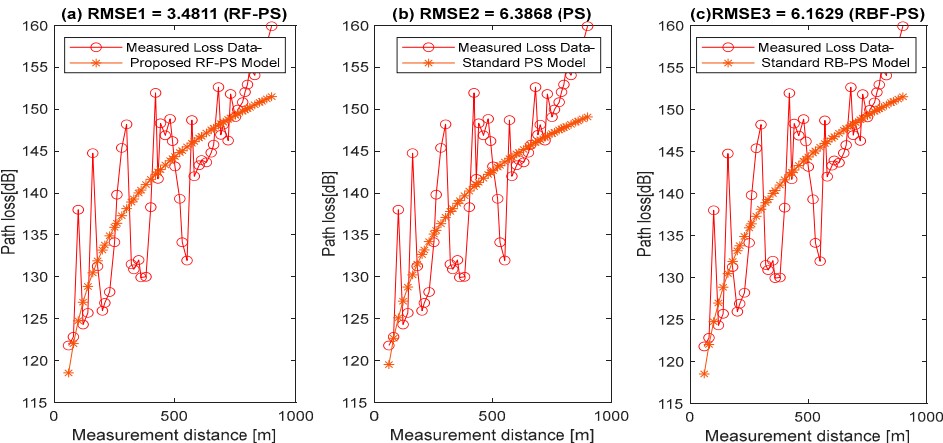

**Figure 18.** Results showing the measured path loss versus distance using the hybrid RF-PS method and two existing standard path loss prediction methods in site 2 of location 4.

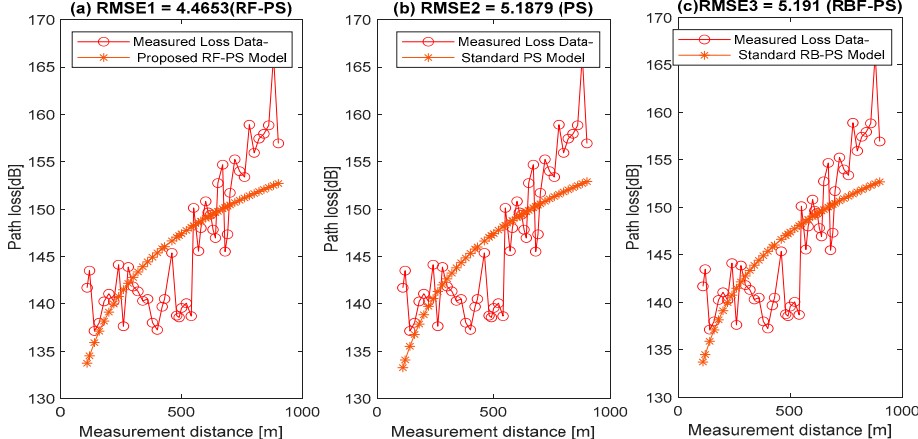

**Figure 19.** Results showing the measured path loss versus distance using the hybrid RF-PS method and two existing standard path loss prediction methods in site 3 of location 4.

The difference between the proposed hybrid RF-PS model and the standard hybrid RBF-PS model is that, while the RBF-PS is merely a combination of two predictive regression approaches, but the proposed hybrid RF-PS model is also a combination of two predictive regression approaches, but with additional feature selection techniques. This enables the

proposed hybrid approach to cater for highly dimensional measured signal path loss data; hence its superior prediction performances over the standard method.

The performance of the enhanced path loss prediction obtained using the proposed method compared with the standard methods is presented, using standard metrics, such as the MAE and R. Lower MAE values obtained using the proposed hybrid RF-PS prediction across all locations are a clear indication of its optimal performance.

Besides the MAE, another very relevant indicator to assess the precision accuracy of the proposed RF-PS method in comparison with measured path loss is the R. Here, R measures the close connection between the predicted and measured path losses. The closer the R-value is to 1, the stronger, healthier, and better the linear connection between the predicted and measured path losses. The linear connection is indicated using a red line in Figures 20–31. The MAE results presented in Table 2 *characterize the magnitude differences between the predicted and the measured path loss* using the proposed hybrid RF-PS prediction and two other existing methods; ordinary PS and combined RBF-PS for sites in Locations 1–4. Table 3 shows the coefficients of the developed path loss model using the proposed hybrid RF-PS approach and other key existing standard approaches.

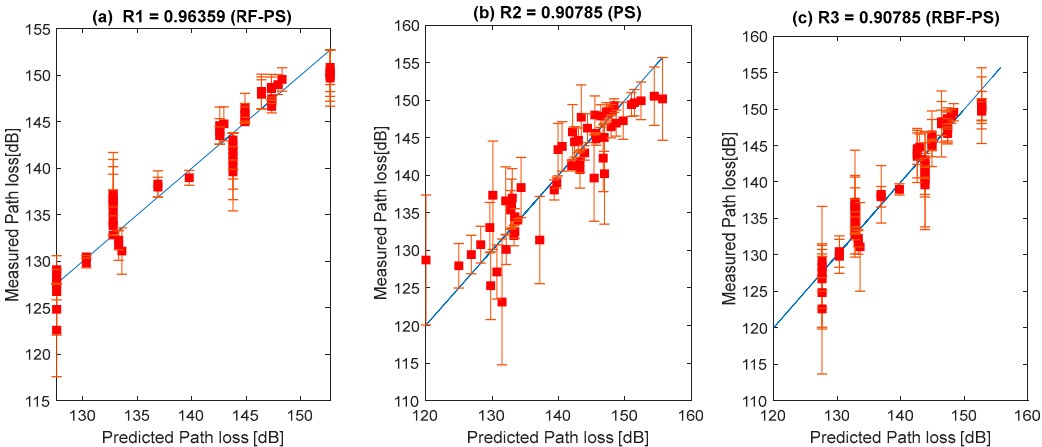

**Figure 20.** Results showing the correlation between the measured path loss and predicted path loss using the hybrid RF-PS prediction method and two existing standard path loss prediction methods in site 1 of location 1.

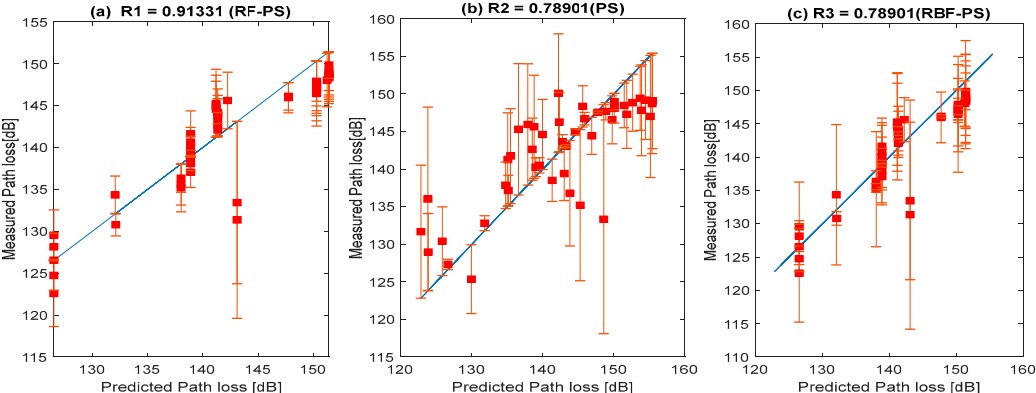

**Figure 21.** Results showing the correlation between the measured path loss and predicted path loss using the hybrid RF-PS prediction method and two existing standard path loss prediction methods in site 2 of location 1.

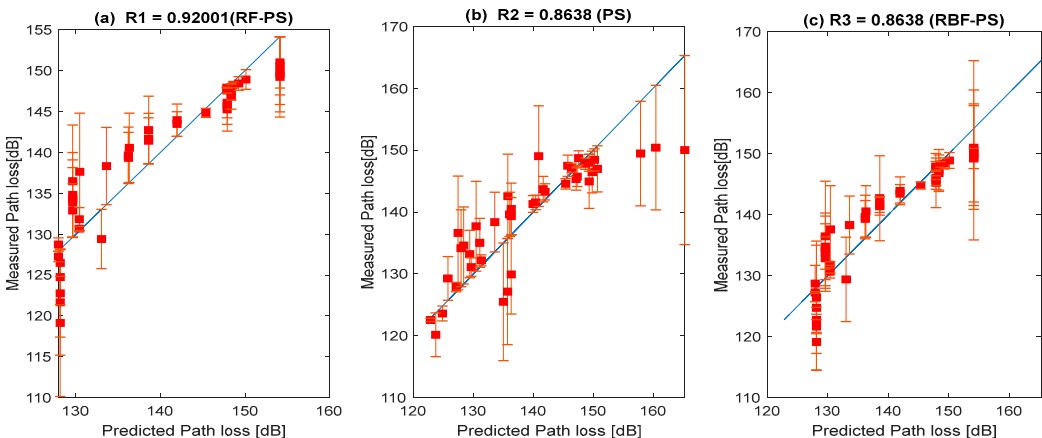

**Figure 22.** Results showing the correlation between the measured path loss and predicted path loss using the hybrid RF-PS prediction method and two existing standard path loss prediction methods in site 3 of location 1.

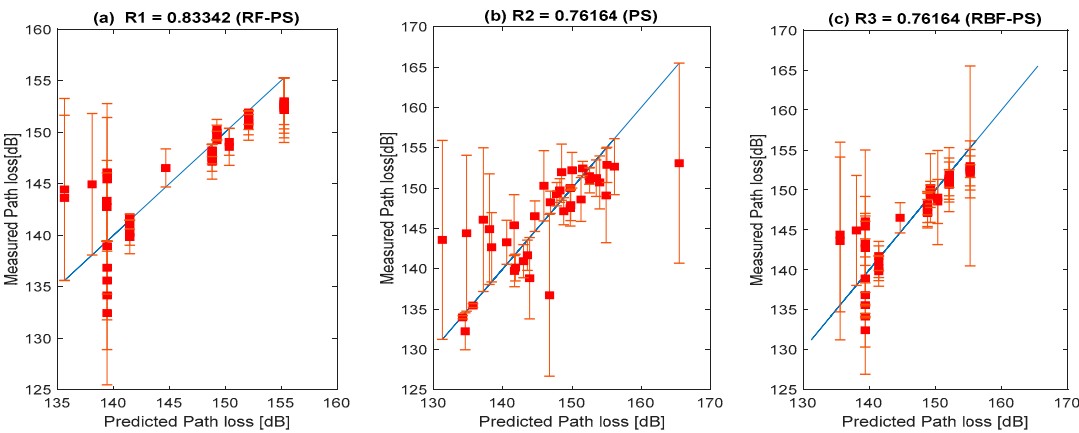

**Figure 23.** Results showing the correlation between the measured path loss and predicted path loss using the hybrid RF-PS prediction method and two existing standard path loss prediction methods in site 1 of location 2.

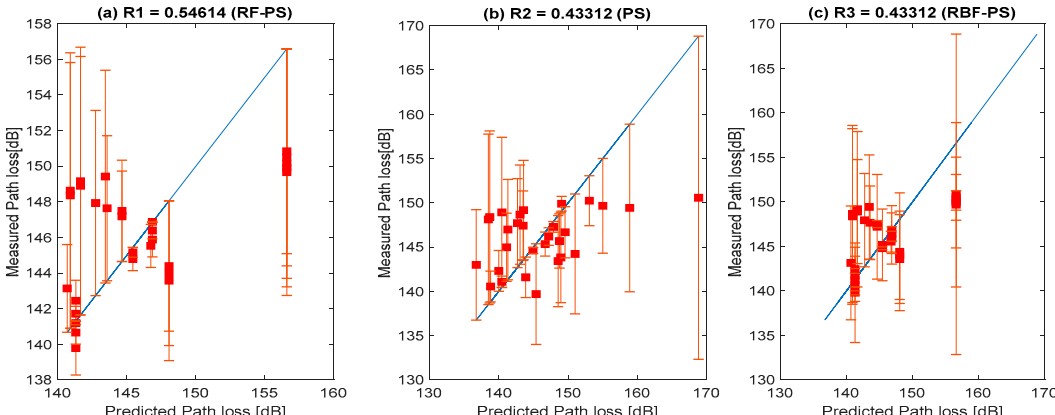

**Figure 24.** Results showing the correlation between the measured path loss and predicted path loss using the hybrid RF-PS prediction method and two existing standard path loss prediction methods in site 2 of location 2.

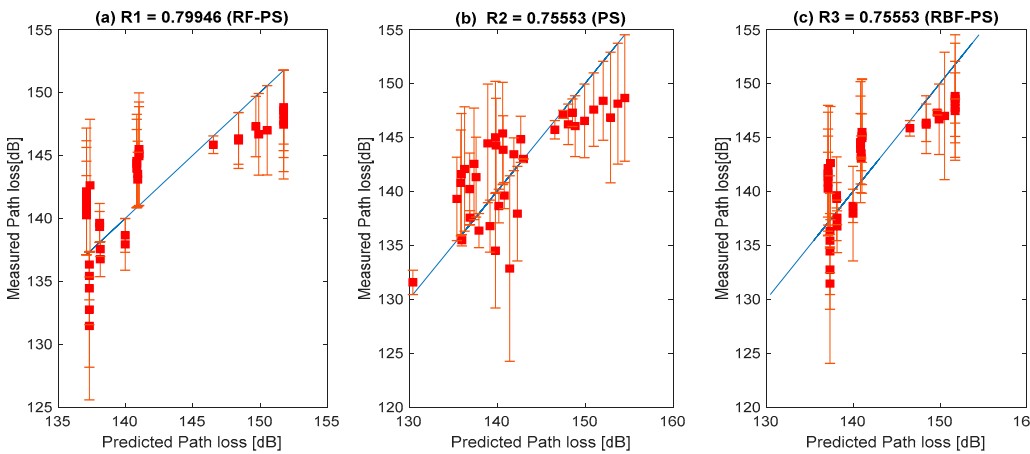

**Figure 25.** Results showing the correlation between the measured path loss and predicted path loss using the hybrid RF-PS prediction method and two existing standard path loss prediction methods in site 3 of location 2.

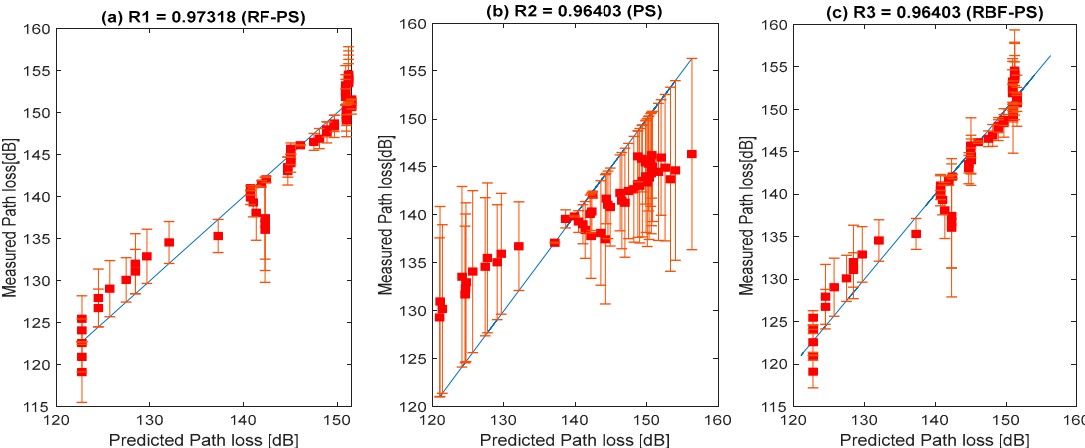

**Figure 26.** Results showing the correlation between the measured path loss and predicted path loss using the hybrid RF-PS prediction method and two existing standard path loss prediction methods in site 1 of location 3.

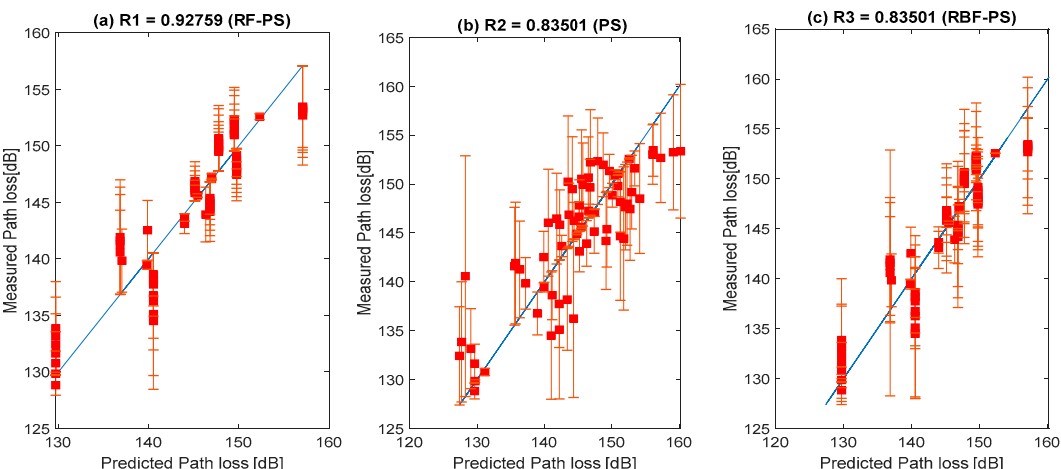

**Figure 27.** Results showing the correlation between the measured path loss and predicted path loss using the hybrid RF-PS prediction method and two existing standard path loss prediction methods in site 2 of location 3.

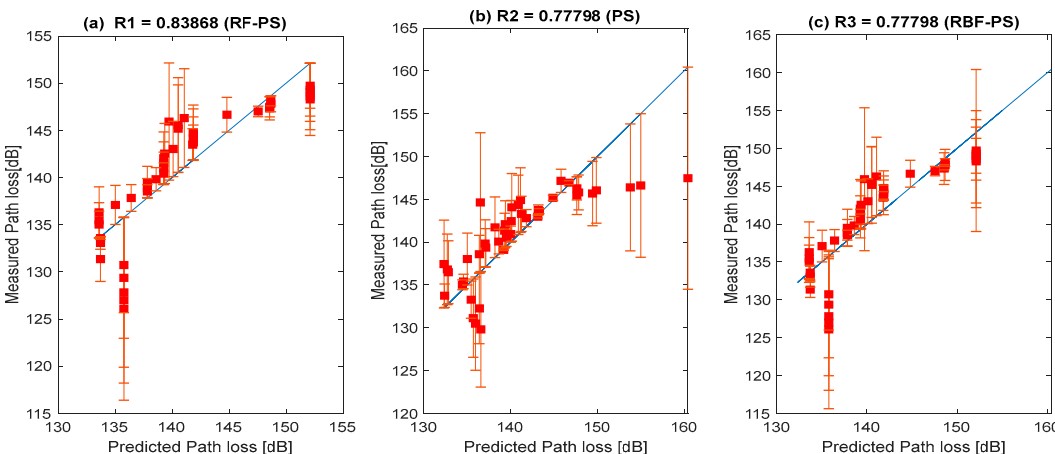

**Figure 28.** Results showing the correlation between the measured path loss and predicted path loss using the hybrid RF-PS prediction method and two existing standard path loss prediction methods in site 3 of location 3.

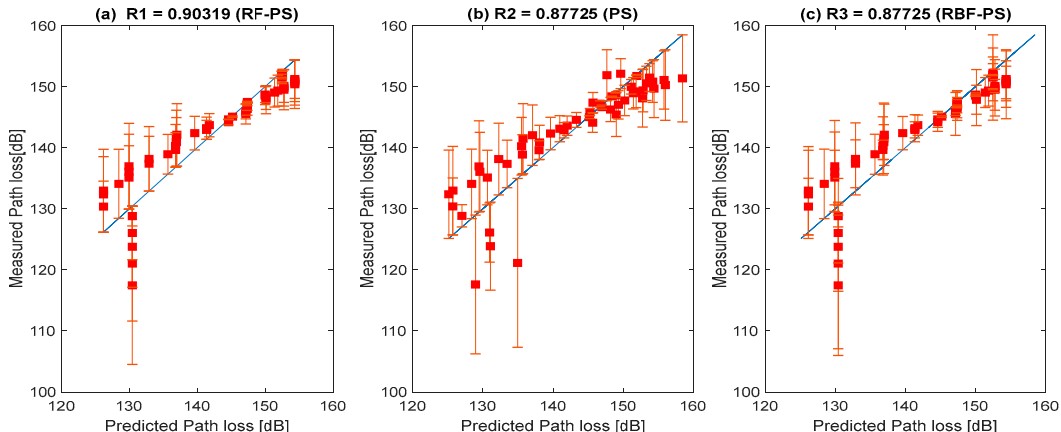

**Figure 29.** Results showing the correlation between the measured path loss and predicted path loss using the hybrid RF-PS prediction method and two existing standard path loss prediction methods in site 1 of location 4.

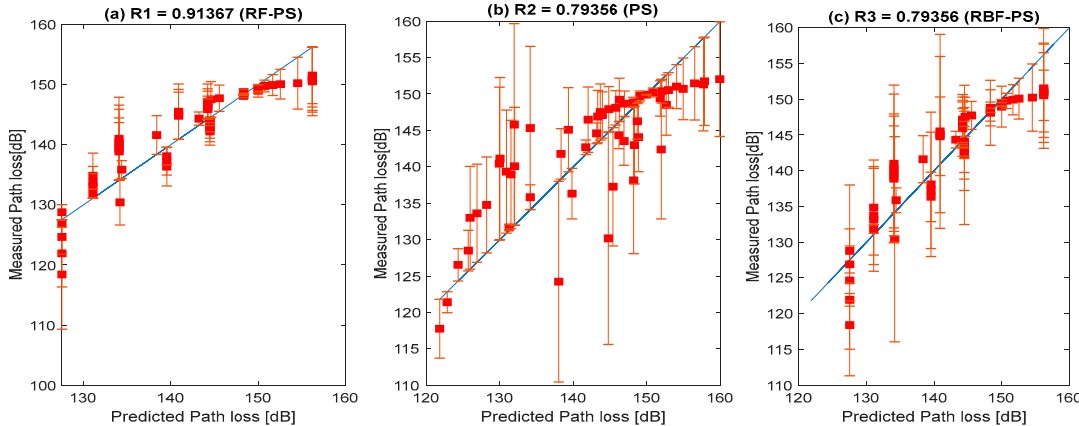

**Figure 30.** Results showing the correlation between the measured path loss and predicted path loss using the hybrid RF-PS prediction method and two existing standard path loss prediction methods in site 2 of location 4.

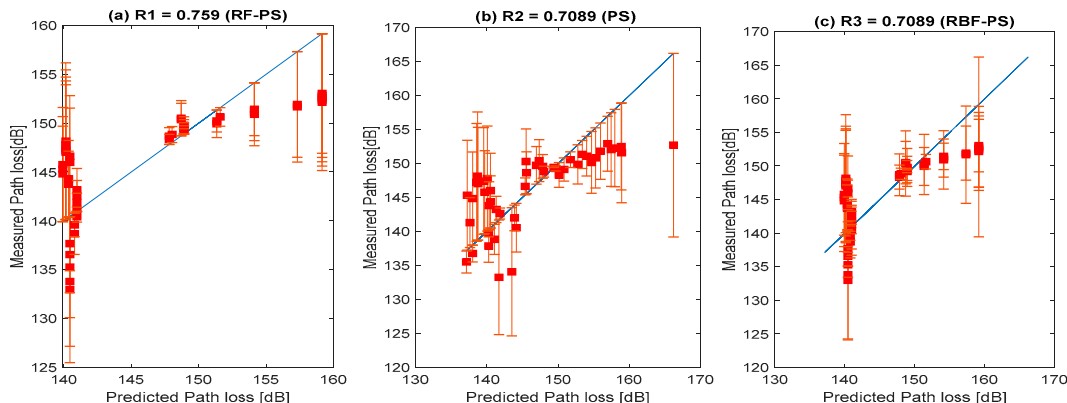

**Figure 31.** Results showing the correlation between the measured path loss and predicted path loss using the hybrid RF-PS prediction method and two existing standard path loss prediction methods in site 3 of location 4.

**Table 2.** Computed MAE values attained with developed path loss model and other key existing standard prediction approaches.

| Location | Site | RF-PS | PS | RBF-PS |
|---|---|---|---|---|
| Asaba | 1 | 2.62 | 3.71 | 3.35 |
| | 2 | 3.39 | 4.47 | 4.45 |
| | 3 | 3.62 | 4.79 | 4.80 |
| Onitsha | 1 | 1.83 | 2.13 | 2.13 |
| | 2 | 2.25 | 3.63 | 3.61 |
| | 3 | 2.82 | 3.00 | 3.03 |
| Awka | 1 | 1.71 | 2.82 | 2.82 |
| | 2 | 2.53 | 4.66 | 4.40 |
| | 3 | 3.07 | 4.02 | 4.02 |
| Agbor | 1 | 3.16 | 3.68 | 3.68 |
| | 2 | 2.83 | 4.97 | 4.97 |
| | 3 | 3.64 | 4.24 | 4.24 |

**Table 3.** The Coefficients of the developed path loss model using the proposed hybrid RF-PS and other existing standard approaches.

| Locations | Proposed Hybrid RF-PS Model Coefficients | | | Standard PS Model Coefficients | | | Standard Hybrid RBF-PS Model Coefficients | | |
|---|---|---|---|---|---|---|---|---|---|
| Parameters | $a_1$ | $a_2$ | $a_3$ | $a_1$ | $a_2$ | $a_3$ | $a_1$ | $a_2$ | $a_3$ |
| Asaba 1 | 831.05 | 21.53 | −217.29 | −102.01 | 45.19 | 305.01 | −339.69 | 6.63 | 136.94 |
| Asaba 2 | 502.79 | 22.54 | −123.06 | −1484.23 | 19.07 | 462.73 | −351.37 | 22.01 | 128.70 |
| Asaba 3 | 78.16 | 22.94 | 1.09 | 1288.98 | 25.43 | −355.39 | 2201.98 | 23.23 | 621.04 |
| Average | 470.66 | 22.37 | −113.08 | −406.09 | 29.90 | 137.45 | 503.64 | 17.29 | 18.47 |
| Onitsha 1 | −235.18 | 39.37 | 87.14 | −497.57 | 39.70 | 157.86 | 1612.03 | 40.01 | 460.11 |
| Onitsha 2 | 13.22 | 27.66 | −19.47 | 1341.66 | 24.50 | 369.47 | 1341.66 | 24.50 | −369.47 |
| Onitsha 3 | 838.77 | 27.66 | 218.94 | 396.71 | 20.78 | 91.01 | −184.69 | 20.99 | 79.10 |
| Average | 198.77 | 27.66 | −47.44 | 413.60 | 28.33 | −100.95 | 923.00 | 28.25 | 56.58 |
| Awka 1 | −2631 | 27.62 | 790.42 | −175.93 | 27.40 | 71.52 | 1763.98 | 27.40 | 496.53 |

**Table 3.** *Cont.*

| Locations | Proposed Hybrid RF-PS Model Coefficients | | | Standard PS Model Coefficients | | | Standard Hybrid RBF-PS Model Coefficients | | |
|---|---|---|---|---|---|---|---|---|---|
| Parameters | $a_1$ | $a_2$ | $a_3$ | $a_1$ | $a_2$ | $a_3$ | $a_1$ | $a_2$ | $a_3$ |
| Awka 2 | −3083.70 | 37.91 | 914.77 | 1525.23 | 22.74 | −422.83 | 796.39 | 26.60 | −212.45 |
| Awka 3 | −330.19 | 30.95 | 113.61 | −676.16 | 32.12 | 214.00 | −6062.90 | 31.80 | 1791.65 |
| Average | −2015.31 | 31.98 | 606.27 | 224.38 | 27.42 | −45.77 | −1177.51 | 28.60 | −360.88 |
| Agbor 1 | 720.85 | 25.11 | −188.97 | −234.84 | 28.26 | 88.47 | 380.36 | 28.19 | −91.62 |
| Agbor 2 | 1048.86 | 21.39 | −281.93 | −690.41 | 28.13 | 222.19 | −780.89 | 28.13 | 248.69 |
| Agbor 3 | −286.99 | 19.80 | 111.53 | −842.39 | 21.53 | 272.82 | −1196.57 | 21.49 | 376.57 |
| Average | 494.24 | 22.10 | −119.61 | −604.47 | 25.97 | 195.83 | −532.37 | 25.94 | 177.88 |

### 3.4. Validation Using the Developed (RF-PS) Optimization Method for Path Loss Prediction in Other Study Locations

Figures 32–35 exhibit the detailed validation results attained with the proposed hybrid RF-PS approach for path loss prediction in four study locations, using four new sites. The results demonstrate that the developed model fitted the measured loss values obtained from the four new locations with high correlation efficiency.

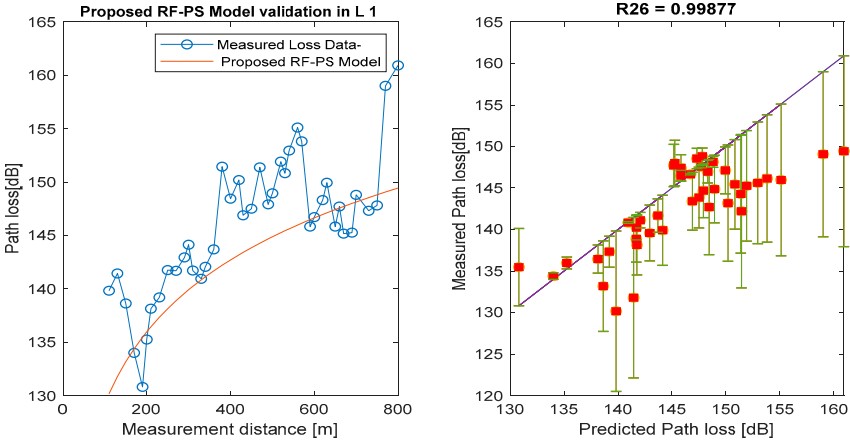

**Figure 32.** Measured path loss and the proposed hybrid RF-PS model validation plot using another site in location 1.

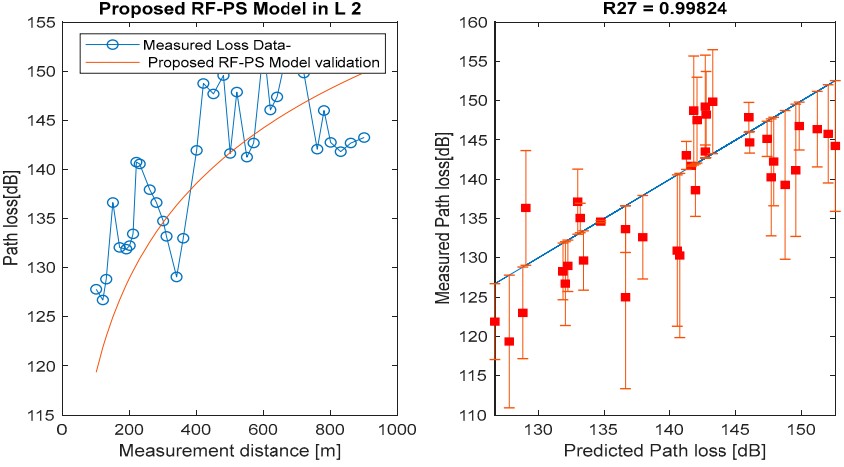

**Figure 33.** Measured path loss and the proposed hybrid RF-PS model validation plot using another site in location 2.

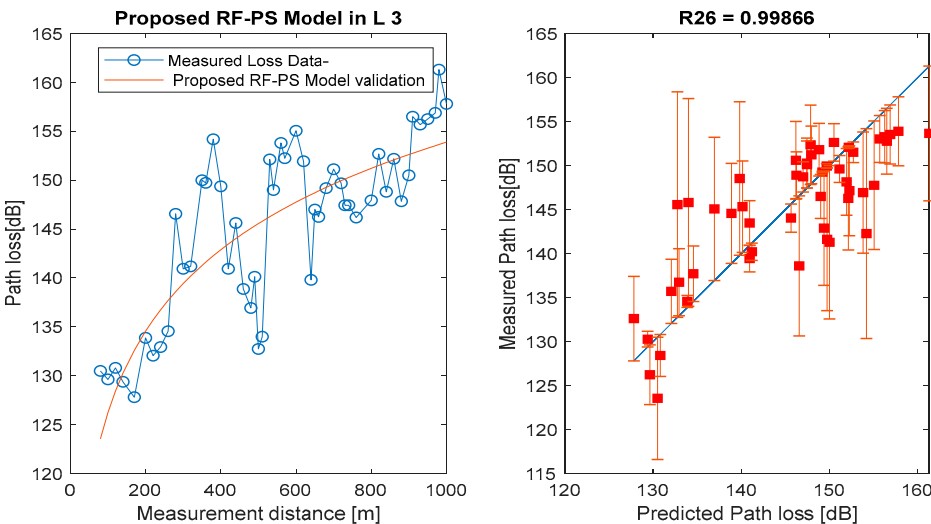

**Figure 34.** Measured path loss and the proposed hybrid RF-PS model validation plot using another site in location 3.

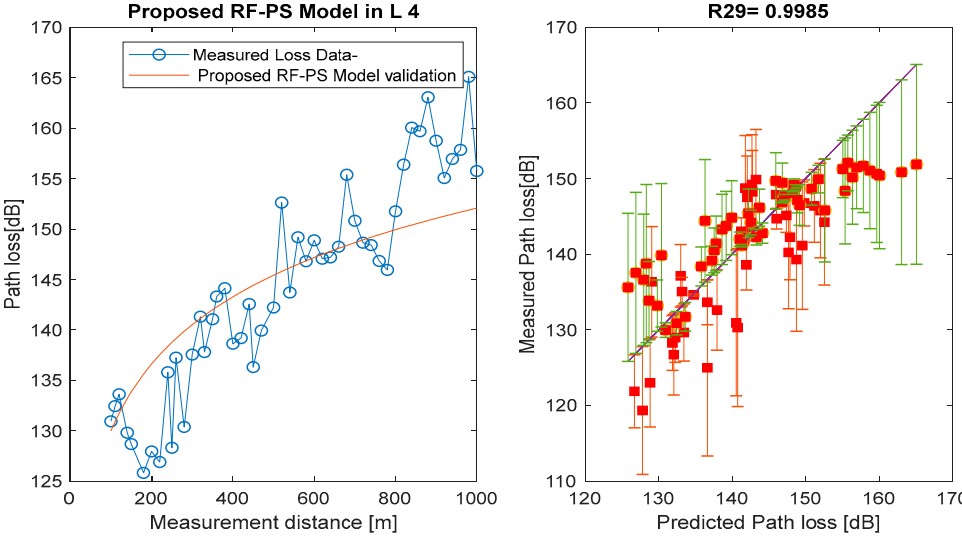

**Figure 35.** Measured path loss and the proposed hybrid RF-PS model prediction validation plot using another site in location 4.

## 4. Conclusions

The rapid growth of cellular-based communication systems has been undergoing viable advancement in terms of multimedia service deployment and mobile subscriptions. The advancements are also being accompanied by the global evolution of other higher broadband cellular communication technologies and deployment infrastructures. In this contribution, a hybrid Random Forest and Particle Swarm Optimization (RF-PS) method was proposed for efficient path loss modeling, leveraging measured path loss data. The proposed RF-PS model exhibits optimal performance in the real-time prognostic analysis of measured path loss data acquired over operational 4G long term evolution (LTE) networks in Nigeria. The proposed RF-PS optimization method showed better prediction performance with lower RMSE values; 3.51–4.33 dB, 2.55–2.98 dB, 3.23–3.54 dB, and 3.11–5.22 dB for locations 1, 2, 3, and 4, respectively. For standard PS and RBF-PS optimization methods, the RMSEs are quite higher; 3.51–4.33 dB, 2.55–2.98 dB, 3.23–3.54 dB, and 3.11–5.22 dB for locations 1, 2, 3, and 4, respectively. These results clearly demonstrate the robustness and superiority of the proposed RF-PS method for path loss prediction over the existing techniques. Future work would focus on the optimization of the projected RF-PS

model to improve its path loss prediction accuracy for application in emerging wireless communication systems.

**Author Contributions:** The manuscript was written through the contributions of all authors. O.R.O., S.A. and J.I. were responsible for the conceptualization of the topic; article gathering and sorting were carried out by O.R.O., S.A. and J.I.; manuscript writing and original drafting and formal analysis were carried out by O.R.O., S.A., J.I., A.L.I., C.-T.L. and C.-C.L.; writing of reviews and editing were carried out by J.I., A.L.I., C.-C.L. and C.-T.L.; J.I. led the overall research activity. All authors have read and agreed to the published version of the manuscript.

**Funding:** This work was supported by the National Science and Technology Council, Taiwan, R.O.C., under contract no.: MOST 110-2410-H-165-001-MY2.

**Data Availability Statement:** The data that support the findings of this paper are available from the corresponding author upon reasonable request.

**Acknowledgments:** The work of Agbotiname Lucky Imoize is supported by the Nigerian Petroleum Technology Development Fund (PTDF) and the German Academic Exchange Service (DAAD) through the Nigerian-German Postgraduate Program under grant 57473408.

**Conflicts of Interest:** The authors declare no conflict of interest related to this work.

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
