# Peer review of "Joint Random Forest and Particle Swarm Optimization for Predictive Pathloss Modeling of Wireless Signals from Cellular Networks"

_futureinternet, doi:10.3390/fi14120373_

Round 1

Reviewer 1 Report

Dear Authors,

the presented work seems to be consistent and logical, but it requires minor improvements. Please consider the following notes:

1. I suggest to move the citation of literature from the end of the paragraph (lines 140-144) and check the line spacing.

2. Equations/formulas are written in a different font size, e.g. (1) and (5) , (6), (7) (lines 145 – 162).

3. There is no explanation of the formulas, e.g. (7) - what is a1, a2, a3 (the explanation appears much later ...).

4. I suggest to improve/rewrite section 2.2.

5. I suggest to change the form of the algorithm presentation (it is worth considering the Latex algorithm template or similar if the article is written in Word) - at least include indentation in loops and conditional statements.

6. Figure 1 is 'narrowed' - I suggest rescaling.

7. Table 1 is 'broken' - move to new page.

8. Please transfer the caption of table 3 in accordance with the journal template.

9. Don't figures 16-27 need a legend? Or at least little more description?

10. Line 416-417: What does a healthier correlation mean for authors?

11. I suggest to be more specific about authors contribution – There is a lack of clear emphasis on what exactly better prediction accuracy means in Section 4 (what is the criterion of goodness? what are the units? how many percent in relation to commonly used models?).

12. I suggest to consider whether table 3 should not be separated into individual coefficients instead of the whole model.

13. I suggest to enrich the part responsible for presenting the current state-of-art and confront with authors solution.

Best regards

Author Response

Dear Respected Reviewer,

Please see attached revision letter. Thanks for your editorial effort.

Best regards,

Chun-Ta Li

Reviewer 2 Report

-the difference between Proposed Hybrid RF-PS model and the Standard Hybrid RBF-PS model needs to be presented

-the role of the RBF Networks (section 2.6) in the proposed model should be shown

-in the section 3. Results and Discussion there is : “Lastly, the fourth part …” p.10 but in the paper there is not analogous section for the fourth part.

-in the conclusion there is the phrase : Finally … and the parameter tuning provides …” It needs more details.

Author Response

(The authors gave the same response as above.)

Reviewer 3 Report

The paper presents a methodology to determine the parameters of a pathloss propagation model using joint Random Forest-Particle Swarm Optimisation (RF PS). The results have show that the proposed method gives better results than other approaches (Particle Swarm or hybrid Radial Basis Function-Particle Swarm Optimization). the design and methodology are well described and can be replicated.

Major comments:

- Report ITU-R M.2412-0 from the International Telecommunication Union defines the Guidelines for evaluation of radio interface technologies for IMT-2020. The proposed propagation model could be additionally benchmarked against the models used in this report, specially for the frequency bands in use by LTE systems in urban, suburban and rural areas.

- No studies were made for indoor propagation scenarios. How does the proposed model perform when used in such environments (which is an environments outside the design coverage)?

- 158: What did the author refers to when they started that "...signal loss in free space attenuates 20dB in value"?

Other comments:

- The presentation of the equation must be improved and their character size should be normalised according to the journal template.

Author Response

(The authors gave the same response as above.)

Round 2

Reviewer 3 Report

The authors have improved the presentation of the paper.

Nevertheless, although the contents of the paper are a valid contribution to the scientific community, I also believe that some of the suggestions proposed remains unanswered:

- ITU-R M.2412-0 is the "de facto" reference used by mobile operators, spectrum regulators and scientists/researchers when implementing celular planning e pathloss studies for several technologies and frequency bands (4G, 5G, etc) for numerous environments (urban, suburban, rural). This can't be considered as "...a path loss model with low precision accuracies when applied in environments other than which they developed.", as stated by the authors, unless they can prove this statement (i.e test it against their results).

Author Response

Dear Respected Reviewer,

Please see attached file. Thanks for your editorial effort.

Best regards,

Chun-Ta Li
